# Remote Sensing Monitoring of Advancing and Surging Glaciers in the Tien Shan, 1990–2019

Sugang Zhou [1], Xiaojun Yao [1,2,*], Dahong Zhang [3], Yuan Zhang [1], Shiyin Liu [4] and Yufang Min [2]

1   College of Geography and Environmental Science, Northwest Normal University, Lanzhou 730070, China; zhousugang@nwnu.edu.cn (S.Z.); zhangyuan@nwnu.edu.cn (Y.Z.)
2   National Cryosphere Desert Data Center, Lanzhou 730000, China; myf@lzb.ac.cn
3   College of Urban and Environmental Science, Northwest University, Xi'an 710027, China; zhangdahong@stumail.nwu.edu.cn
4   Institute of International Rivers and Eco-security, Yunnan University, Kunming 650091, China; shiyin.liu@ynu.edu.cn
*   Correspondence: xj_yao@nwnu.edu.cn; Tel.: +86-0931-797-1161

**Abstract:** The advancing of glaciers is a manifestation of dynamic glacial instability. Glaciers in the Tien Shan region, especially in the Central Tien Shan, show instability, and advancing glaciers have been recently detected. In this study, we used Landsat TM/ETM+/OLI remote sensing images to identify glaciers in the Tien Shan region from 1990 to 2019 and found that 48 glaciers advanced. Among them, thirty-four glaciers exhibited terminal advances, and 14 glaciers experienced advances on the tributary or trunk. Ten of the glaciers experiencing terminal advances have been identified as surging glaciers. These 48 glaciers are distributed in the western part of the Halik and Kungey Mountain Ranges in the Central Tien Shan, and Fergana Mountains in the Western Tien Shan, indicating that the Tien Shan is also one of the regions where advancing and surging glaciers are active. From 1990 to 2019, a total of 169 times advances occurred on 34 terminal advancing glaciers in the Tien Shan region; the highest number of advancing and surging of glaciers occurred in July (26 and 14 times, respectively). With reference to the existing literature and the present study, the surge cycle in the Tien Shan is longer than that in other regions at high latitudes in Asia, lasting about 35–60 years. Surging glaciers in the Tien Shan region may be affected by a combination of thermal and hydrological control. An increase in temperature and precipitation drives surging glaciers, but the change mechanism is still difficult to explain based on changes in a single climate variable, such as temperature or precipitation.

**Keywords:** advancing glaciers; surging glaciers; remote sensing monitoring; Tien Shan



## 1. Introduction

The cryosphere, one of the five major spheres of the climate system, is highly sensitive to climate change [1]. With the background of global warming, changes in the cryosphere and their effects have attracted widespread attention [2,3], becoming one of the hotspots of current global change research [3]. Alpine glaciers, an important part of the cryosphere, serve as frozen reservoirs of freshwater and as sensitive indicators of climate change [4]. Since the 1980s, with global warming, glaciers in western China have generally been in mass deficit, showing a trend of retreat and thinning [5]. Local climatic conditions and topography cause glaciers in some areas to show different change trends [6]. In recent years, the "Karakorum anomaly" phenomenon, characterized by the relative stability and even advance of glaciers in the Karakoram Range, Pamirs and West Kunlun Mountains, has been focus for many scholars [7–9].

A surging glacier refers to the rapid movement, periodically, of a glacier in a relatively short period (2–3a) [10]. Surging glaciers suddenly accelerate their movements in the surge phase, causing a rapid transfer of glacial mass so that the ice is suddenly redistributed

while the total glacial mass remains unchanged, demonstrating obvious changes in surface features such as medial moraine folds, fragmentation of ice surfaces, and the rapid advance of the glacier termini [10,11]. Global reports and studies of glacial surges have focused on glaciers in Svalbard and East Greenland [12], the Yukon in Canada, Alaska [13], the Karakoram Range [14–18], Pamirs [19,20], West Kunlun Mountains [21,22], and the Caucasus Mountains [23]. The rapid movement of glaciers during the surge period can destroy ranches, roads, bridges, villages, and hydroelectric facilities by a surge within a short time [24,25], and can also trigger other disasters, such as glacial lake outbursts, debris flows, etc. [26,27], causing a huge threat to the infrastructure and the safety of residents' lives and property in the downstream area. Therefore, research on the features of distribution, the occurrence mechanism, possible damage, and prevention mechanism of surging glaciers have attracted scholars and local governments.

The Tien Shan region, which hosts a large number of modern glaciers, has attracted numerous studies related to the glacial mass balance [28,29], glacial changes [30–32], and ice volume [33]. Few records and studies of advancing or surging glaciers have been done. In 1982, Dolgushin was the first to discover evidence that a tributary of the Wukuer Glacier in the Muzart Valley in the Tien Shan region had surged [34]. The authors of [35] analyzed the evolution of the Northern Inylchek Glacier and Glacier Lake (Upper Lake Merzbacher) during the surging period, using remote sensing images. The authors of [36] summarized the surging glaciers of the Central Tien Shan from 1960 to 2014 based on the literature and remote sensing images. The above studies are only related to parts of the Tien Shan or individual surging glaciers; however, comprehensive and continuous observation of advancing or surging glaciers in the Tien Shan were lacking, especially concerning their current status.

In this study, we aim to (1) identify glaciers that may be advancing and surging in the Tien Shan region by comparing changes in the surfaces and termini of glaciers based on remote sensing imagery acquired in 1990–2019; (2) determine the timing and frequency of glacial advancing and surging; (3) attempt to explore the mechanism of advancing and surging glaciers, the surge cycle, and the response of glaciers to climate change in this region; and (4) analyze the potential disaster risk to infrastructure in downstream areas from these glaciers in this region. The goal is to improve our understanding of the characteristics of glacial changes and assist in glacial disaster prevention in the Tien Shan region.

## 2. Materials and Methods

### 2.1. Study Area

The Tien Shan Range (40°–45°N, 69°–95°E) in the hinterland of the Eurasian continent stretches from Xingxing Gorge in Hami, Xinjiang, China in the east to the Kyzylkum Desert in Uzbekistan in the west. This range is connected with the Pamirs in the south and the Altai Mountains in the north and east, spanning four countries—China, Kazakhstan, Kyrgyzstan and Uzbekistan—and is the largest independent latitudinal mountain system in the world (Figure 1). The Tien Shan Range is about 2500 km long from east to west and 250–350 km wide on average from north to south, with the widest point reaching more than 800 km. The average elevation is about 4000 m with the highest peak, Tomur Peak (7435.3 m above sea level), located in China. Situated in the middle temperate zone with a mountainous continental climate, the Tien Shan Range has an obvious vertical zonality. The annual precipitation in high elevation areas can reach 400–800 mm, creating climatic conditions very conducive to the formation and development of glaciers and permafrost in the Tien Shan, whereas the annual precipitation at low elevations is about 100–200 mm [37–39]. Known as the "Water Tower of Central Asia", glaciers in the Tien Shan feed numerous rivers, such as the Chu, Syr, Ili and Tarim, and their meltwater provide important water supplies supporting the existence of surrounding rivers and oases [39].

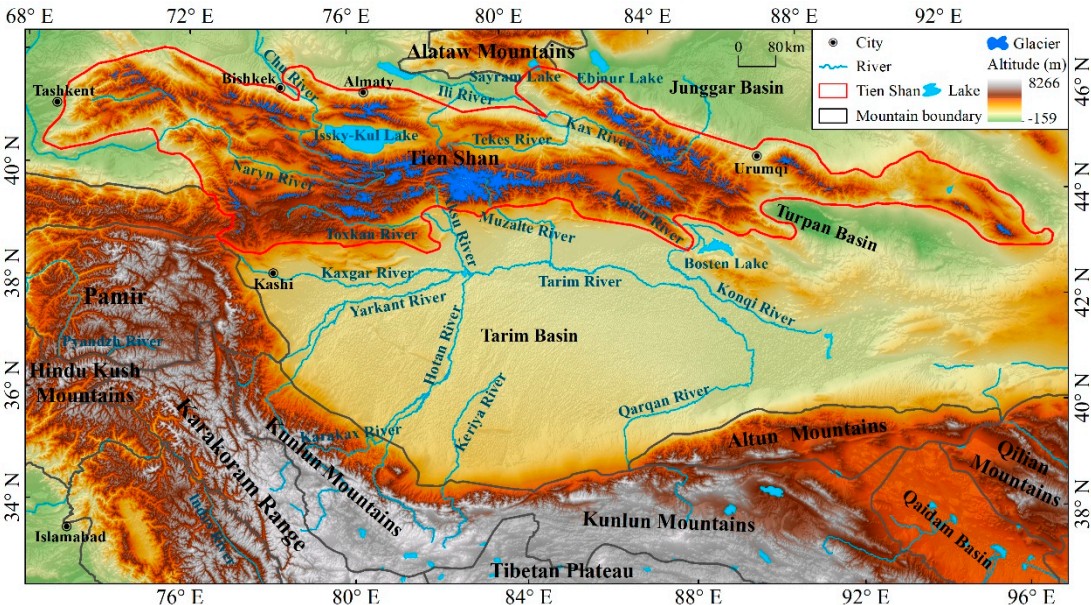

**Figure 1.** Digital elevation model (ASTER GDEM V2) of the study area shows elevation of the Tien Shan Range spanning from northern Xinjiang, China to parts of Kazakhstan, Kyrgyzstan, and Uzbekistan (political boundaries are not shown) and the Tarim Basin and surrounding mountain ranges.

According to the latest Randolph Glacier Inventory V6.0 [40], 13,998 glaciers exist in the Tien Shan with a total area of 11,864 km$^2$. Among them, seven glaciers cover areas greater than 100 km$^2$ with a total area of 1602.39 km$^2$, accounting for 0.05% and 13.5% of the total number and area of glaciers in the Tien Shan, respectively. The South Inylchek Glacier in northeastern Kyrgyzstan is the largest glacier in the Tien Shan, covering an area of 373.92 km$^2$ followed by the Tuomuer Glacier in China with an area of 358.25 km$^2$.

*2.2. Data Sources*

Outlines of glaciers need to be extracted at different times to analyze the dynamics of glaciers. Two authoritative sets of basic glacial data were selected for the delineation of glacier outlines in different years. Glacier data for the territory of China came from the Second Glacier Inventory Dataset of China V1.0 [41], downloaded from the National Tibetan Plateau Data Center (http://data.tpdc.ac.cn/en/, accessed on 1 December 2020). Glacier data for other countries came from the Randolph Glacier Inventory V6.0 [40], downloaded from the Global Land Ice Measurements from Space (GLIMS) (http://www.glims.org/RGI/, accessed on 14 December 2020).

The changes in the glacier terminal and surface morphology provided the main basis for identifying advancing and surging glaciers along with the base for extracted glacier boundaries. A total of 702 remote sensing images were used in this study (Figure 2), including Level-1 products of the Landsat Thematic Mapper (TM), Enhanced Thematic Mapper (ETM+), and Operational Land Imager (OLI) downloaded from the United States Geological Survey (USGS) (https://earthexplorer.usgs.gov, accessed on 5 November 2020). To reduce the influence of snow and clouds on the extraction of glacier boundary, images without clouds and less snow were selected as much as possible. All the images had been treated with radiation, geometric, and topographical corrections based on digital elevation model (DEM) data. In addition, the visible (30 m) and panchromatic (15 m) bands of OLI data were fused to form images with a spatial resolution of 15 m in this study.

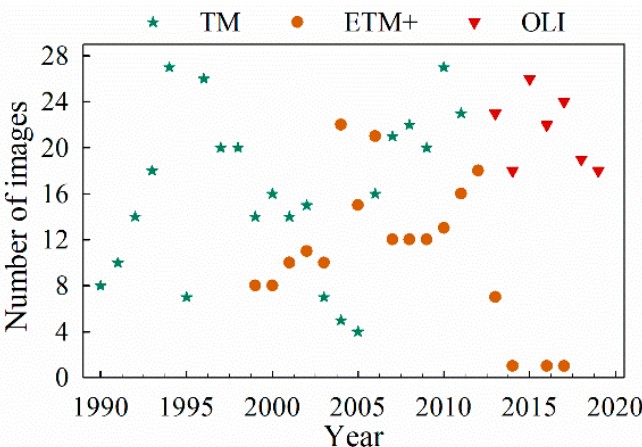

**Figure 2.** Time series of Landsat thematic mapper (TM), enhanced thematic mapper (ETM+), and operational land imager (OLI) remote sensing images in the Tien Shan Range show the number of images acquired for each year 1990–2019.

The DEM data used to extract of ridge line and calculate glacier length is ASTER GDEM V2 product with a spatial resolution of 30 m, obtained from the Geospatial Data Cloud site (http://www.gscloud.cn, accessed on 11 January 2021) of the Computer Network Information Center of the Chinese Academy of Sciences. The DEMs used to calculate the surface elevations of glaciers were Shuttle Radar Topography Mission (SRTM) and ASTER Digital Elevation Model (AST14DEM) V003, which were downloaded from the USGS and NASA Earth Data (https://earthdata.nasa.gov/, accessed on 5 April 2021), respectively. Among them, SRTM data were acquired in 2 scenes (n41_e079_1arc_v3, n42_e079_1arc_v3), AST14DEM data were acquired in 3 scenes (AST14DMO_00310032011 055711_24296_DEM, AST14DMO_00310312004055638_24312_DEM, AST14DMO_0030512 2007053845_7174_DEM), both with spatial resolution of 30 m.

In the study, a new climate reanalysis product of the European Centre for Medium-Range Weather Forecasts (ECMWF), ERA5-Land Climate Reanalysis, released in 2017, was used to analyze the response of advancing and surging glaciers to climate change based on the temperature 2 m above the ground and precipitation data. The ERA5-Land Climate Reanalysis dataset is the 5th generation product of the Global Climate Reanalysis Data with an hourly temporal resolution and a horizontal resolution of $0.1° \times 0.1°$, which was downloaded from the ECMWF website (https://www.ecmwf.int/, accessed on 24 December 2020).

*2.3. Methods*

2.3.1. Glacial Boundary Extraction

Scholars have adopted various methods to extract glacial boundaries, including manual visual interpretation, the band ratio threshold method, the normalized snow cover index method, and the band ratio threshold method combined with visual interpretation [42–46]. Studies have shown that the band ratio threshold method based on multispectral remote sensing images combined with visual interpretation is relatively accurate in extracting glacial boundaries [47]. In this paper, we used the band ratio combined with the manual revision method to extract glacier boundaries in the Tien Shan region. Firstly, binary images of the glacier area were obtained using the band ratio (Landsat TM\ETM+: Band3/Band5, Landsat OLI: Band4/Band6), and after repeated trials, the threshold value was selected in the range of 1.8~2.0. Meanwhile, the smaller image elements were removed using a median filter (window size of 3 × 3, units are image cell size) (Figures 3a–c and 4a–c). Then, the binary image was converted to vector data, and the glacial boundaries were manually revised in conjunction with the glacier inventory data and Google Earth images (Figure 3d–e). With reference to the extraction method of the Second China Glacier Inventory [47], the boundaries of debris-covered glaciers were manually extracted based

on characteristics of the image color and texture features, distribution of glacial lakes, hydrological features of glacial termini, and topography on both sides of each glacial and water system feature (Figure 4d–e). Finally, the ASTER GDEM data were used to extract the ridgeline, and the individual glacier vector boundary was obtained by splitting the revised glacier boundary with ridgeline (Figures 3f and 4f).

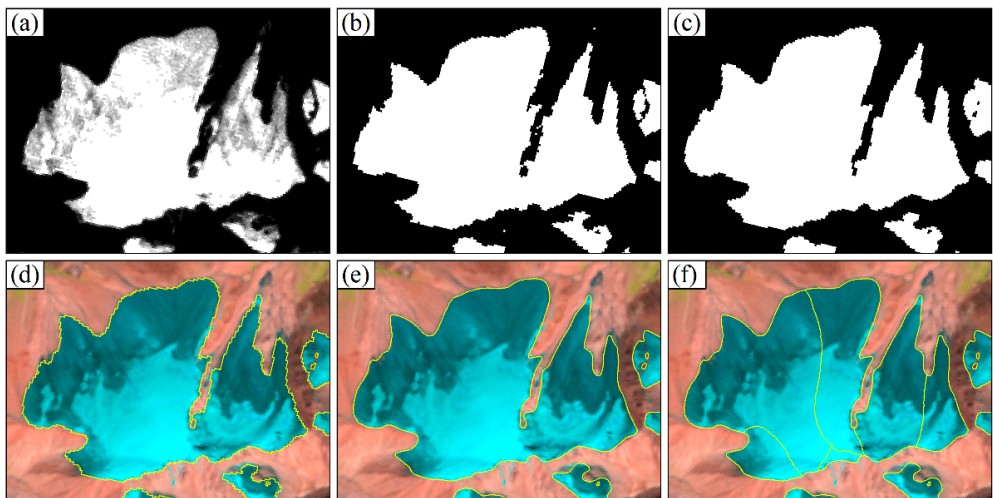

**Figure 3.** Bare ice boundary extraction ((**a**) Band3/Band5 ratio image, (**b**) image with a threshold of 1.8, (**c**) image after using the median filter, (**d**) glacier outline in vector format, (**e**) image after manual revision, and (**f**) individual glacier outline; (**d**–**f**) Landsat Thematic Mapper (TM) image (Bands 5, 4, and 3); the image is LT51480312006240IKR00).

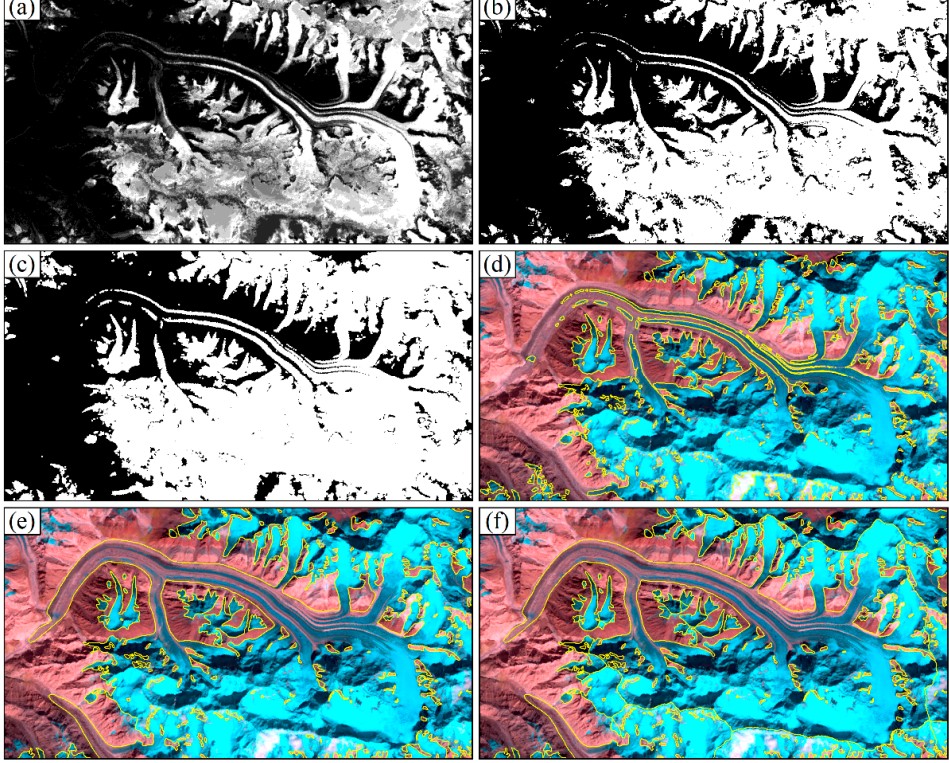

**Figure 4.** Debris-covered glacier boundary extraction ((**a**) Band3/Band5 ratio image, (**b**) image with a threshold of 1.8, (**c**) image after using the median filter, (**d**) glacier outline in vector format, (**e**) image after manual revision, and (**f**) individual glacier outline; (**d**–**f**) Landsat Thematic Mapper (TM) image (Bands 5, 4, and 3); the image is LT51470312007236IKR00).

### 2.3.2. Glacier Length Extraction

Glacier length refers to the maximum distance along the glacier axis, namely, the maximum length of the main stream line [48]. Glacier length is an important parameter used for the reconstruction of climate change, assessing glacial ice volume, construction of a glacier dynamics model, and the prediction of future changes in glaciers. There are many methods for the extraction of glacier length, and two main ones are currently available: the glacier mainstream method and glacier centerline method. The former adopts hydrological analysis models to obtain the glacier collection waterline used to extract the glacier length [49,50], while the latter extracts glacier length by calculating the centerline from the highest to the lowest point of a glacier [51–53]. By comparing the advantages and disadvantages of the glacier length extraction methods proposed by different scholars, the authors of [54,55] developed an automatic method for extracting the glacier centerline for different types of glaciers, which was used in the present study to extract the lengths of glaciers in the Tien Shan. The basic principle is that the highest and lowest points on the glacier outlines are obtained based on the DEM data, and the extracted highest and lowest points are used to split the glacier outlines. Using the segmented glacier outlines data as input, the glacier is split into two regions using Euclidean distance, i.e., each region is the set of points closest to each glacier outline segment, and the common edge of the region is the central axis, i.e., the glacier centerline [51].

The accuracy of glacier length extraction depends on the accuracy of the glacier outlines and the quality of the DEM data, although the latter has a negligible effect on the glacier length [54,55]. Therefore, the accuracy of the glacier length only depends on the error caused by the spatial resolution of the remote sensing images, which is calculated by Equation (1):

$$\varepsilon = \left(1 - \frac{\lambda}{L}\right) \times 100\% \tag{1}$$

where $\varepsilon$ is the accuracy of glacier length extraction (%), $L$ is the glacier length (m), and $\lambda$ is the spatial resolution of the remote sensing images (30 m, 30 m, and 15 m for Landsat TM, ETM+, and OLI, respectively). The calculation results showed that the overall accuracy of the glacier length measurements in the Tien Shan reached 99.42%.

### 2.3.3. Glacier Surface Elevation Calculation

The AST14DEM product is generated using bands 3N (nadir-viewing) and 3B (backward-viewing) of an (ASTER Level 1A) image acquired by the Visible and Near Infrared (VNIR) sensor. The band 3 stereo pair is acquired in the spectral range of 0.78 and 0.86 microns with a base-to-height ratio of 0.6 and an intersection angle of 27.7 degrees. The DEM data are produced by the Sensor Information Laboratory Corporation ASTER DEM/Ortho (SILCAST) software developed by LP DAAC, currently in version 3.0. The software uses the ephemeris and attitude data derived from both the ASTER instrument and the Terra spacecraft platform to produce DEMs and orthorectification data without the need for ground control points or reliance on external global DEMs at 30-arc-second resolution (GTOPO30), the outputs are geoid height corrected and waterbodies are automatically detected in this new version. We also selected SRTM data because Aster DEM data are subject to image quality. We calculated the change in glacier surface elevation after co-registration processing of all data. The error of the glacier surface elevation calculation is subject to the accuracy of the DEM itself and the error of co-registration [56,57]. In this study, we used the error calculation method proposed by authors of [58] to evaluate the error of glacier surface elevation calculation:

$$e = \sqrt{SE^2 + MED^2} \tag{2}$$

$$SE = \frac{STDV_{\text{no glac}}}{\sqrt{n}} \tag{3}$$

where $e$ is the error of elevation change, and $SE$ and $MED$ are the standard error and mean elevation difference in the non-glacial area, respectively. $STDV_{\text{no glac}}$ is the standard

deviation of the non-glacial region, and $n$ is the number of image elements contained in the non-glacial region. Referring to the literature of [58], we chose an auto-correlation distance of 600 m for the ASTER DEM with a spatial resolution of 30m.

### 2.3.4. Identification of Advancing and Surging Glaciers

Glacier retreat, advance, and surge are often intermixed in the same area, increasing the difficulty of accurately identifying surging glaciers [59]. The rapid advancement phenomenon of the glacial terminus is the most significant feature for the identification of surging glaciers based on the remote sensing technique [56]. Therefore, the identification process of this study is shown in Figure 5. If a glacial terminus advanced more than 60 m during the entire study period, the glacier was considered to have advanced (Figure 5a) [60]. If the latter phase advanced more than 150 m over the previous phase glacier terminus, the glacier was preliminarily confirmed to be a surging glacier (Figure 5a) [10]. The surging glacier also shows its surface with moraine folds, surface crevasses, a broken surface and other phenomena; in addition, the glacier terminus is leaf- and/or drop-shaped (Figure 5b,c) [10,15,34]. The changes in the glacial surface elevation can be used as a basis for identifying surging glaciers (Figure 5d); for example, the surface elevation of the reservoir area decreases significantly, and the surface elevation of the receiving area increases significantly after a glacial surge [59]. Here, we defined the surface advancing glaciers as a phenomenon in which the glacier's terminus does not change significantly, but the glacial surface (of a trunk or tributary glacier) rises and moves forward to a new location. Figure 5e shows a diagram of a glacier before and after surface advance.

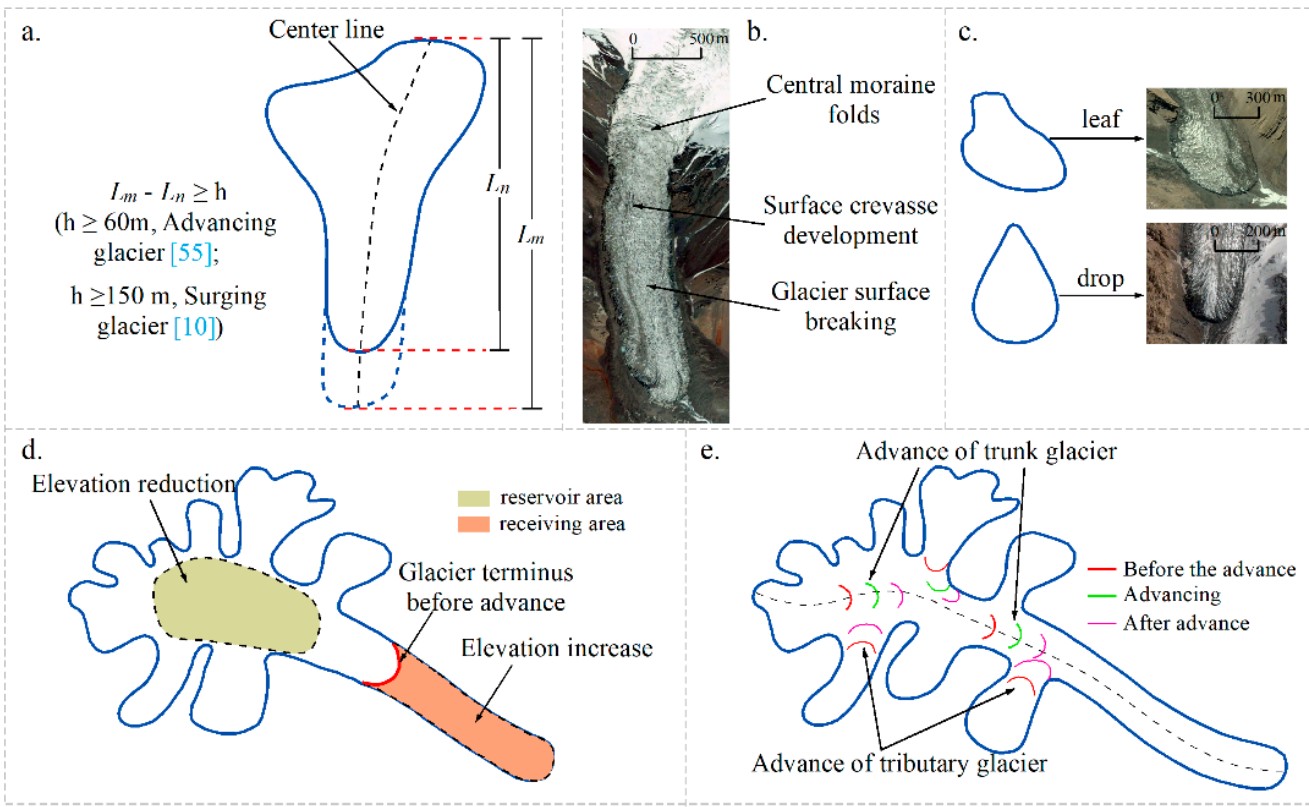

**Figure 5.** Diagrams of advancing and surging glaciers: (**a**) drawing of terminal changes in advancing and surging glaciers; (**b**) aerial photo of surface characteristics of surging glaciers; (**c**) aerial photo of terminal shapes (leaf and drop shapes) of surging glaciers; (**d**) drawing illustrating changes in surface elevation of surging glaciers; (**e**) drawing illustrating glacier forms with surface advance. Note: h—length of glacier advance; $L_m$—length of glacier after advance; $L_n$—length of glacier before advance. The images in figure (**b**,**c**) are Google Earth historical images.

## 3. Results

Based on Landsat TM/ETM+/OLI images and glacier inventory data, 48 glaciers were found to have advanced to various extent by comparing changes of each glacier from 1990 to 2019 in the Tien Shan Range (Figure 6). Among them, 34 glacial termini advanced significantly, and 14 glacier surfaces advanced. According to the method of identification of advancing and surging glaciers, 10 of the 34 terminal advances of glaciers were identified as surging glaciers, and the others were advancing glaciers. Table 1 lists the basic parameters of these glaciers. The 48 advancing glaciers identified here are mainly located in the western part of Halik Mountain and Kungey Mountain in the Central Tien Shan, and in Fergana Mountain in the Western Tien Shan. No advancing glaciers were identified in the eastern part of the Central and Eastern Tien Shan.

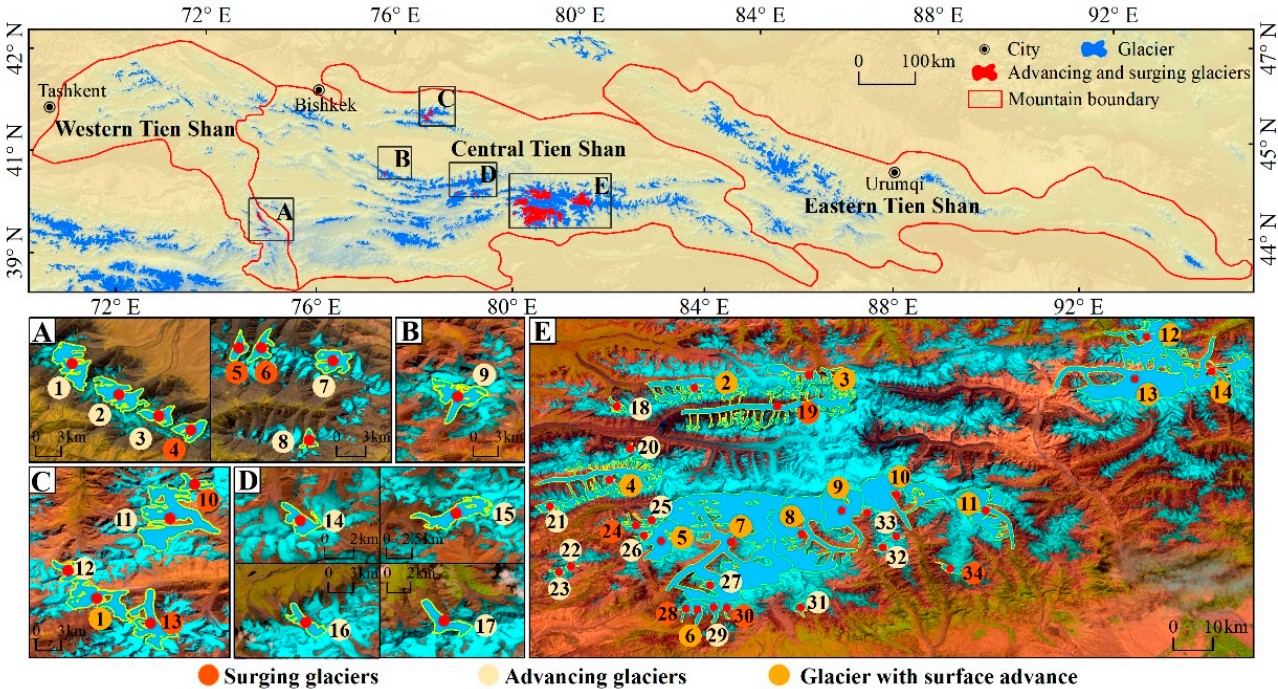

**Figure 6.** Distribution of advancing and surging glaciers in the western, central, and eastern Tien Shan region (background: the main figure shows an ASTER GDEM V2 product of the Tien Shan Range; (**A–E**) Landsat Operational Land Imager (OLI) image (Bands 6, 5, and 2); the images are LC81500322016266LGN02, LC81490312016259LGN01, LC81490302016259LGN01, LC81480312018225LGN00, and LC81470312016181LGN01).

### 3.1. Terminal Advancing Glaciers

From 1990 to 2019, the surface characteristics of the 24 advancing glaciers did not change obviously, but their termini all advanced (Figure 7) by less than 1 km. Among them, six glaciers (Nos. 2 (right branch), 7, 16, 26, 27, 32) experienced terminal advances between 400 and 900 m. Eleven glaciers (Nos. 1, 2 (Left branch), 3, 8, 9, 12, 15, 20, 22, 25, 30) experienced terminal advances between 200 and 400 m. Eight glaciers (Nos. 11, 14, 17, 18, 21, 23, 29, 33) experienced terminal advances between 100 and 200 m. The termini of the left and right tributary of the No. 2 glacier both advanced by 352 m and 846 m, respectively, for periods of 4 and 13 years, respectively. The terminus of the Bogatyr Glacier advanced 165 m, whereas its branch glacier advanced 457 m. Ice fractures occurred and surface debris cover reformed in some glaciers; supraglacial lakes disappeared and the terminal hydrology was altered in others.



**Table 1.** Basic parameters of advancing and surging glaciers in the Tien Shan Range (calculated from Randolph Glacier Inventory V6.0; advancing and surging glaciers).

| No. | Global Land Ice Measurements from Space ID number | Name | Area (km²) | Mean Elevation (m) | Mean Slope (°) | Orientation | Advancing or Surging |
| --- | --- | --- | --- | --- | --- | --- | --- |
| 1 | G074329E40728N | - | 9.86 | 4107 | 18.26 | E | advancing |
| 2 | G074380E40695N | - | 9.25 | 4096 | 18.45 | NE | advancing |
| 3 | G074422E40672N | - | 3.81 | 4105 | 17.86 | NE | advancing |
| 4 | G074456E40657N | - | 3.99 | 4135 | 18.31 | NE | surging |
| 5 | G074571E40560N | - | 2.87 | 4246 | 17.93 | N | surging |
| 6 | G074550E40560N | - | 2.17 | 4056 | 14.93 | N | surging |
| 7 | G074646E40548N | - | 4.46 | 4349 | 16.11 | NE | advancing |
| 8 | G074616E40475N | - | 1.34 | 4250 | 16.25 | N | advancing |
| 9 | G076673E41922N | - | 9.89 | 4192 | 14.04 | S | advancing |
| 10 | G077289E43078N | Shokalsky Glacier | 2.82 | 3922 | 22.36 | NW | surging |
| 11 | G077248E43020N | Bogatyr Glacier | 24.49 | 4040 | 12.40 | E | advancing |
| 12 | G077152E42985N | - | 3.05 | 3973 | 19.51 | NW | advancing |
| 13 | G077241E42927N | South Jangyryk Glacier | 10.90 | 4030 | 14.80 | N | surging |
| 14 | G078877E42099N | - | 1.96 | 4304 | 20.00 | N | advancing |
| 15 | G078676E42013N | - | 6.72 | 4431 | 17.58 | W | advancing |
| 16 | G078307E41952N | Bezymyanny Glacier | 4.50 | 4298 | 19.02 | N | advancing |
| 17 | G078370E42156N | - | 3.11 | 4017 | 21.80 | NW | advancing |
| 18 | G079736E42242N | - | 5.12 | 4178 | 23.85 | N | advancing |
| 19 | G080097E42241N | Northern Inylchek Glacier | 112.88 | 4309 | 16.90 | W | surging |
| 20 | G079762E42144N | - | 1.87 | 4260 | 32.89 | N | advancing |
| 21 | G079584E42014N | - | 9.99 | 4398 | 22.57 | NW | advancing |
| 22 | G079632E41877N | - | 2.17 | 4226 | 35.67 | E | advancing |
| 23 | G079604E41864N | - | 3.57 | 4446 | 36.44 | N | advancing |
| 24 | G079781E41970N | Samoilowich Glacier | 6.72 | 4343 | 18.85 | NW | surging |
| 25 | G079816E41982N | East of Samoilowich Glacier | 6.74 | 4371 | 16.53 | NW | advancing |
| 26 | G079799E41948N | - | 3.38 | 4388 | 21.32 | W | advancing |
| 27 | G079950E41834N | - | 5.52 | 4436 | 33.60 | W | advancing |
| 28 | G079894E41780N | Qingbingtan Glacier | 6.60 | 4589 | 29.09 | S | surging |
| 29 | G079962E41786N | Qinbingtan No. 74 Glacier | 8.64 | 4528 | 20.53 | S | advancing |
| 30 | G079989E41783N | Kekeer Glacier | 5.01 | 4492 | 23.86 | S | surging |
| 31 | G080157E41783N | - | 5.77 | 4329 | 22.35 | S | advancing |
| 32 | G080350E41918N | - | 2.73 | 4605 | 29.14 | SW | advancing |
| 33 | G080375E41947N | - | 1.72 | 4359 | 22.85 | E | advancing |
| 34 | G080499E41872N | - | 5.71 | 4323 | 21.91 | NW | Surging |

A total of 104 advances occurred on 24 advancing glaciers in the Tien Shan region from 1990 to 2019, and all of them occurred two or more times (Figure 8). Among them, the right branch of the G074380E40695N glacier experienced the largest number of advances (10 times). In terms of month (Figure 8a), the advance phenomenon occurred in every month, except November, in the Tien Shan Range; it mainly occurred in July and August, followed by September (26, 23, and 13 times, respectively), and only once in December and April, respectively. In terms of year (Figure 8b), glacial advances occurred mainly in 2013 (10 times) in the Tien Shan region, followed by 1997 and 2015 (both eight times). No advances occurred in 1990, 1991, 1995, 1998, and 2012. The greatest number of advances occurred from 2010 to 2019 (42 times), followed by 2000 to 2010 (35 times); the least occurred from 1990 to 2000 (27 times), with a gradual increase in recent years. From the perspective of the duration of advancing, glaciers in the Tien Shan region advanced without obvious regularity. The shortest period of advancement was about 2 years and the longest one was 17 years. However, this number ranged from 3 to 7 years for most glaciers.

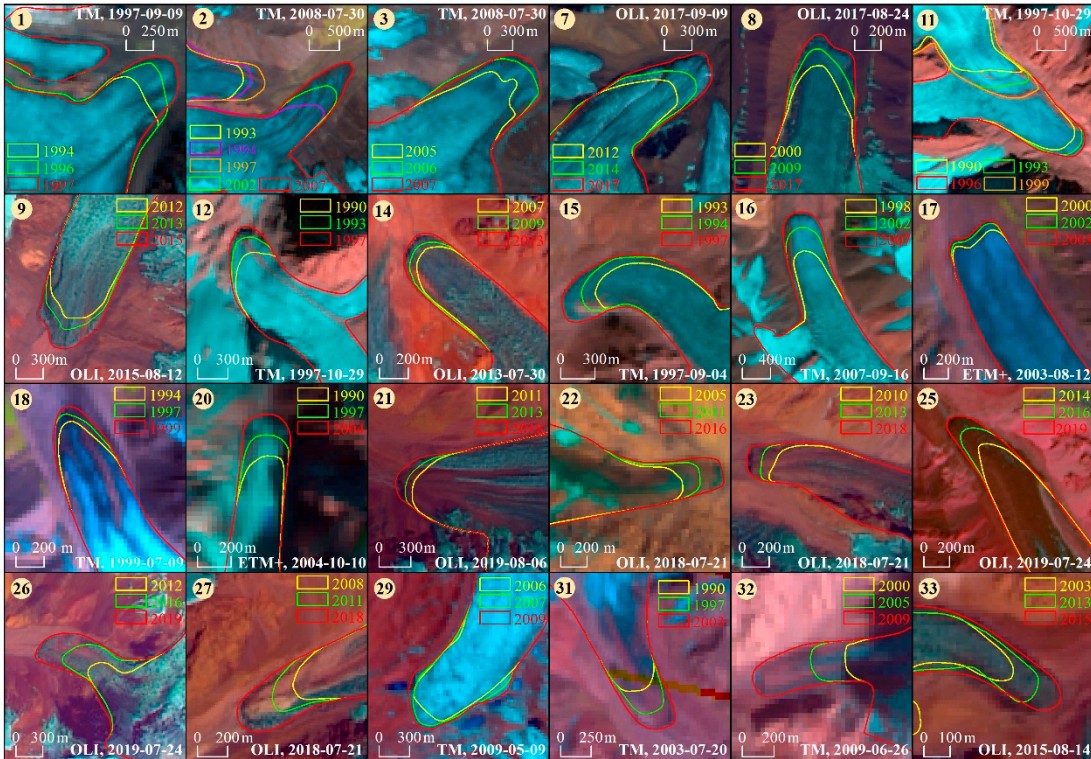

**Figure 7.** Changes in the termini of advancing glaciers in the Tien Shan Range. Figures shows the beginning (yellow), middle (green), and end (red) glacial boundaries in various years and with remote sensing images of advancing glaciers (refer Table 1 for glacier numbers): glaciers 1, 2, 3, 7, 8, 11, 9, 12, 14, 15, 16, 17, 18, 20, 21, 22, 23, 25, 26, 27, 29, 31, 22, and 33 from different dates acquired using Landsat Thematic Mapper (TM), Enhanced Thematic Mapper (ETM+), and Operational Land Imager (OLI) imagery.

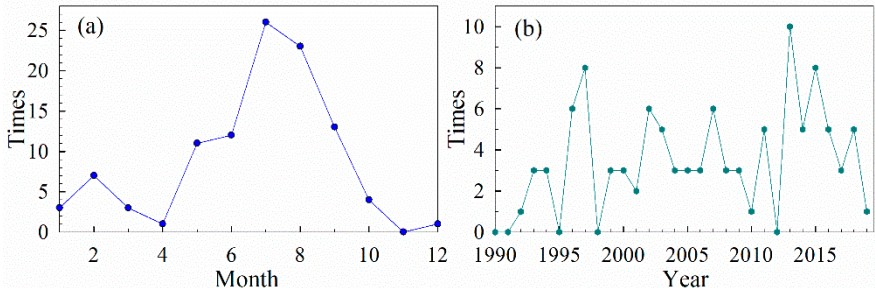

**Figure 8.** Number of advances of advancing glaciers occurring in different (**a**) months and (**b**) years.

### 3.2. Surging Glaciers

There were 10 surging glaciers in the Tien Shan region: Northern Inylchek, Samoilowich, Shokalsky, and South Jangyryk along with glacier Nos. 4, 5, 6, 28, 30, 34. The termini of all these glaciers experienced rapid advances, whereas the glacier surfaces also changed noticeably, showing broken surfaces, folded medial moraines, and changes in ice tongue shapes (Figure 9, Table 2). When the terminal and surface morphology of surging glaciers were compared, using remote sensing images and Google earth imagery, the Northern Inylchek Glacier advanced 3571 m at the terminus from 1995 to 1997. From October 1996 to February 1997, the glacier terminus advanced rapidly by about 2404 m (15.82 m/d) in a few months, and the glacial lake, Upper Lake Merzbacher, was almost completely covered by the rapid advance of the glacier terminus. The authors of [61] discovered that the reservoir zone of this glacier thinned by about 100 m and the receiving zone thickened by about 150 m from 1975 to 1999; the receiving zone continuously thinned in subsequent years.

While the Samoilowich Glacier advanced a total of 2799 m between 1999 and 2006, its terminus advanced slowly from 1999 to 2001 with an advance of 383 m (191.5 m/a) over two years. After 2001, the advancement of glacial termini accelerated greatly, with the termini advancing 720 m from 2001 to 2002, 1138 m from 2002 to 2003, and 509 m from 2003 to 2004, totaling 2416 m (483.2 m/a) from 2001 to 2006. The shape of the terminus changed markedly, showing a leaf shape. Meanwhile, we discovered that the average thinning of the glacier reservoir area was about −33 ±1.13 m, and the maximum thinning about −59 ±1.13 m; the average thickening of the receiving area was about 51 ±1.13 m, and the maximum thickening about 128 ± 1.13 m from 2000 to 2007 (Figure 10a). The No. 34 glacier advanced by 1673 m with the longest surge duration (19 years) and the highest number of advances (18 times) from 1990 to 2009 and surged almost every year. For this glacier, the glacial termini are drop-shaped due to topographical constraints. The No. 6 glacier advanced a total of 907 m between 2002 and 2010, with the glacier terminus advancing rapidly by 384 m (192 m/a) from 2005 to 2007. We discovered that the average thinning of the glacier reservoir area was about −44 ± 2.34 m, and the maximum thinning was about −69 ± 2.34 m; the average thickening of the receiving area was about 62 ± 2.34 m, and the maximum thickening was about 177 ± 2.34 m from 2004 to 2011 (Figure 10b). The South Jangyryk Glacier and the No. 5 and No. 30 glaciers advanced 700–1000 m during the active period, with a surge duration ranging from 2 to 9 years. The Shokalsky Glacier and No. 4 and No. 28 glaciers advanced about 500 m, with a duration of active advances of 3–6 years.

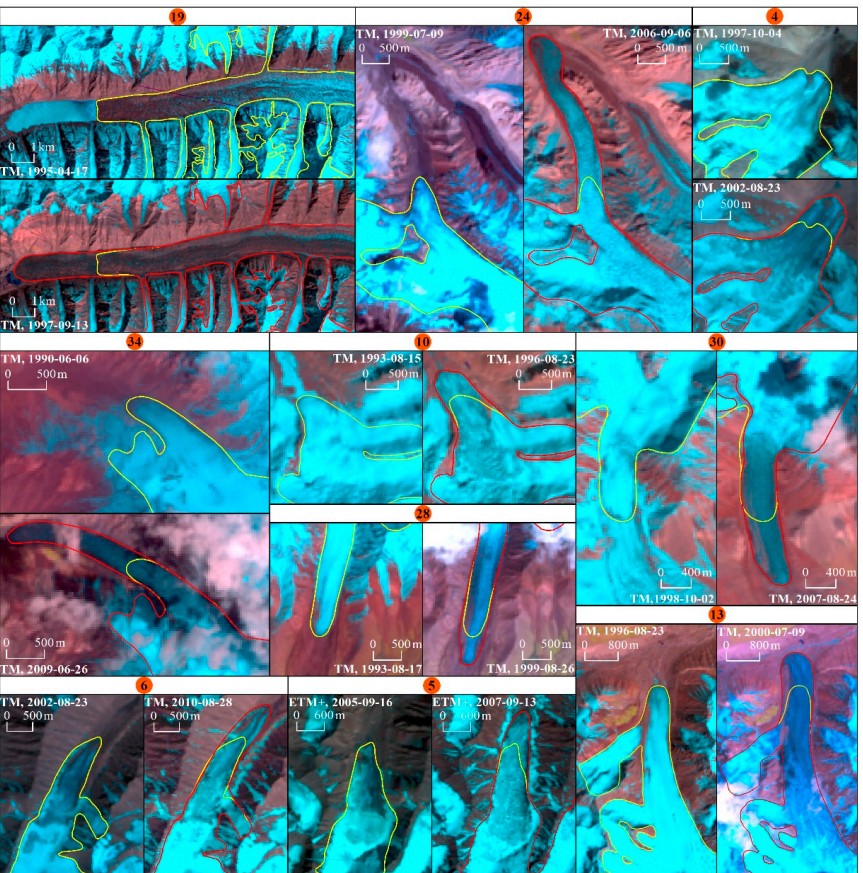

**Figure 9.** Surging glaciers in the Tien Shan Range. Figure shows glacial boundaries at the beginning and end of a surge using remote sensing images acquired at the end of the surging glacier (refer to Table 1 for glacier numbers): glaciers 19, 24, 4, 34, 10, 30, 28, 13, 6, and 5 acquired using Landsat Thematic Mapper (TM) and Enhanced Thematic Mapper (ETM+) imagery.

**Table 2.** Detailed information on each surging glacier in the Tien Shan region including evidence of surge event, total length of advance (m), surge initiation (year), and surge termination (year).

| No. | Evidence of Surge Events | Total Length of Advance (m) | Surge Initiation (year) | Surge Termination (year) |
|---|---|---|---|---|
| 4 | Terminal advanced 208 m from 1998 to 1999; glacial surface breaking. | 511 | 1997 | 2002 |
| 5 | Terminal advanced 355 m/a; glacial surface breaking. | 710 | 2005 | 2007 |
| 6 | Terminal advanced 348 m from 2005 to 2007; glacial surface breaking. | 907 | 2002 | 2010 |
| 10 | Terminal advanced 370 m from 1994 to 1996; glacial surface breaking. | 441 | 1993 | 1996 |
| 13 | Glacier termini show clear movement in remote sensing images; glacial surface breaking. | 729 | 1996 | 2000 |
| 19 | Rapid terminal advance of 3457 m from 1996 to 1997; glacial lake disappears; glacial surface breaking. The elevation of the glacial surface is obviously thickened and thinned. | 3571 | 1995 | 1997 |
| 24 | Rapid terminal advance of 1010 m from 2003 to 2004; glacial surface breaking; surface crevasse development; terminus is leaf shaped. The elevation of the glacier surface is obviously thickened and thinned. | 2799 | 1999 | 2006 |
| 28 | Terminal advanced 181 m from 1994 to 1995; glacial surface breaking; surface crevasse development. | 537 | 1993 | 1999 |
| 30 | Terminal advanced 198 m from 1998 to 1999; glacial surface breaking; surface crevasse development. | 870 | 1998 | 2007 |
| 34 | Terminal advanced 189 m from 2001 to 2002; glacier termini show clear movement; glacial surface breaking; surface crevasse development; terminus is drop-shaped. | 1673 | 1990 | 2009 |

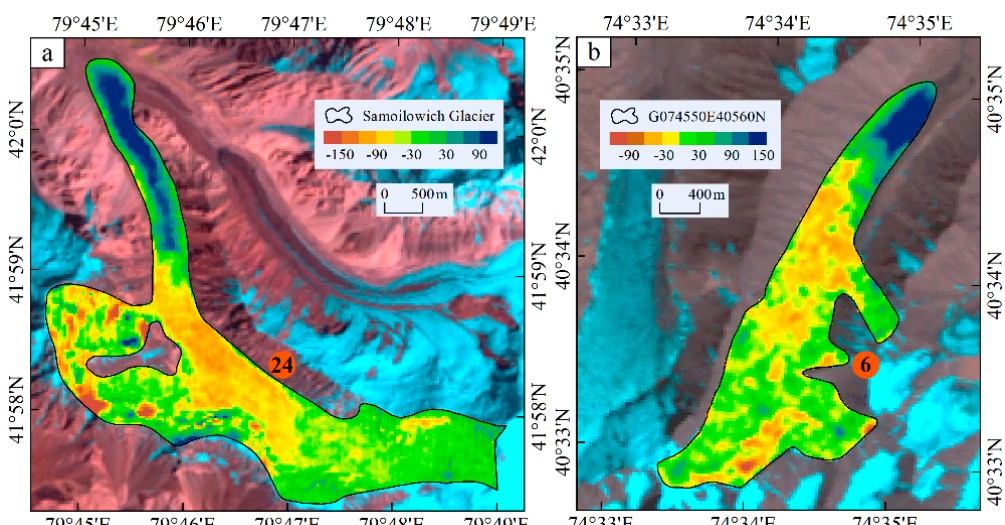

**Figure 10.** The figure shows (**a**) glacier thickness change (m) of glacier 24 from 2000 to 2007 and (**b**) glacier thickness change of glacier 6 from 2004 to 2011 (background: (**a**,**b**) Landsat Thematic Mapper (TM) image (Bands 5, 4, and 3); the images are LT51470311996222ISP00, and LT51500322008212KHC01).

A total of 65 glacial advances occurred to 10 surge glaciers in the Tien Shan region from 1990 to 2019. In terms of month (Figure 11a), the surge of glaciers occurred in any month except December in the Tien Shan region, with surges mainly concentrated in July, August, September (14, 11, and 10 times, respectively), and January (once). In terms of year (Figure 11b), the glaciers mainly surged in 2000 and 2002 (six times) in the Tien Shan region,

followed by 1996, 1997, 1998, and 1999 (five times each); only one surge occurred in 1991, 1992, and 2010, whereas no advances occurred after 2010. There was a slight difference in the number of surge events between 2000–2010 (34 times) and 1990–2000 (31 times) in the Tien Shan region, with surge events mainly occurring before 2010.

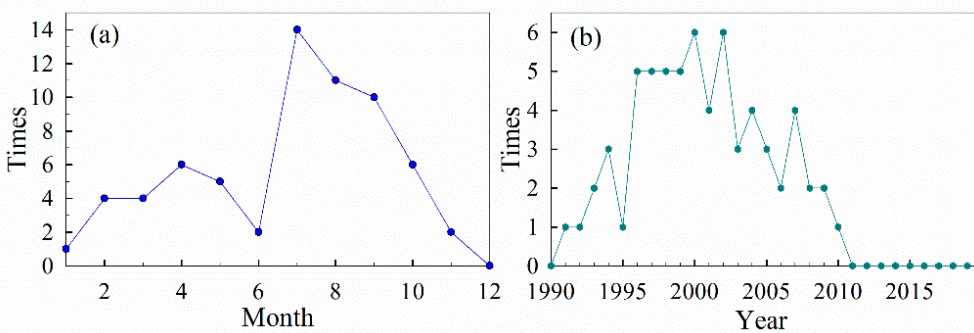

**Figure 11.** Number of advances of surging glaciers occurred in different (**a**) months and (**b**) years.

### 3.3. Surface Advancing Glaciers

Based on remote sensing imagery, in addition to identifying the terminal advances of glaciers in the Tien Shan, 14 surface tributary and trunk advance glaciers were identified (Figure 12). The glacier surfaces were covered by debris, but the glacier termini were not advanced obviously, which means these glaciers are in a stable or even slowly retreating state. Some glacial surface motion information is relatively difficult to identify from remote sensing images in areas with a high coverage ratio of clouds and snow, so some very high-quality images were selected. The results show that the surfaces of all 14 glaciers have advanced to different extents from 1990 to 2019; some glaciers experienced advances in both the tributary and trunk glaciers (Table 3). Interestingly, six of the seven tributary glaciers of the Mushketov Glacier advanced, but the trunk glacier terminal retreated year by year. Advance of the Ayilangsu Glacier occurred at five places, with advance of the tributary glacier A1 beginning in 1992, which caused compression of the trunk glacier, and narrowed its width by about 426 m. After that, the advance of the trunk glacier forced the tributary glaciers to move forward with the trunk glacier, so the surface width of the trunk glacier increased by about 157 m by 2005. Branch Glacier A1 started advancing again in 2002, and the second advance also squeezed the trunk glacier, narrowing the surface width by about 214 m as of 2019. The West Qiongtailan Glacier advanced in four places; the two trunk glaciers advanced more than 2 km, and they were in a state of advancement throughout the study period (1990–2019). Its tributary glaciers began to advance in 1990, and the upper part of the tributary glaciers advanced a second time in 2003. Trunk and tributary glacial advances occurred mainly in the central part of the Central Tien Shan, which indicates that the glaciers in this region are in an unstable state. Therefore, glacial surge events are likely to occur in the future. Some errors are involved in documenting the exact timing of glacier movement due to the influence of the temporal resolution and the image quality of the remote sensing images, but this is inevitable. In the future, more time-intensive and better-quality images will be required to analyze the velocity of glacial surfaces; further research needs to be done in this field.

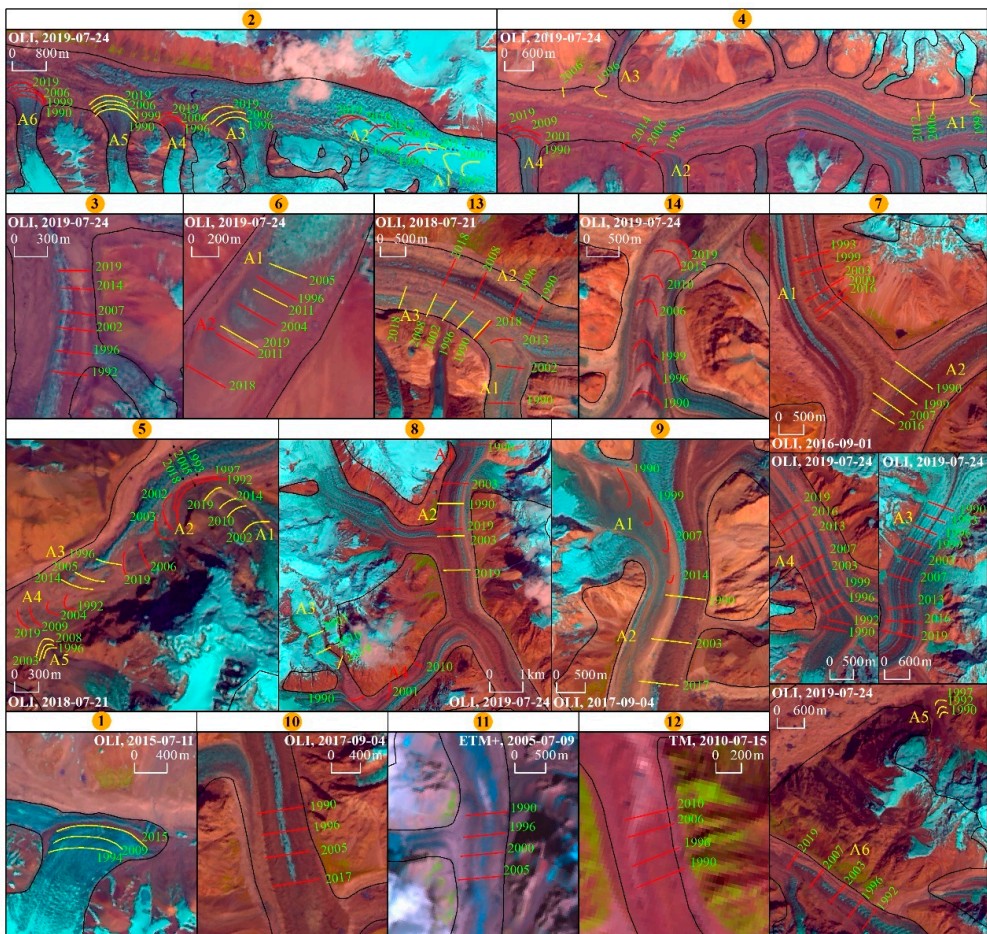

**Figure 12.** Timing of advancing glacial surface of the Tien Shan region. Figure shows locations for the year of advancing surface tributaries, trunk glaciers, labeled on remote sensing images of the most recent period of advancing glaciers in this sequence (including tributaries/trunks as follows) (refer Table 1 for glacier numbers): glaciers 2 (A1–6), 4 (A1–4), 3, 6 (A1–2), 13 (A1–3), 14, 7 (A1–6), 5 (A1–5), 8 (A1–3), 9 (A1–2), 1, 10, 11, and 12 using Operational Land Imager (OLI) imagery, except for glaciers 11 and 12, which used Enhanced Thematic Mapper (ETM+) and Landsat Thematic Mapper (TM) imagery, respectively.

**Table 3.** Detailed information on the advance of tributary and trunk glaciers in the Tien Shan Range (surface advancing glaciers).

| No. | Global Land Ice Measurements from Space ID Number | Name | Characteristics/Description |
|---|---|---|---|
| 1 | G077183E42955N | - | Tributary glacier: 360 m |
| 2 | G079906E42284N | Mushketov Glacier | Tributary glacier: A1, 997 m; A2, 1874 m; A3, 492 m; A4, 261 m; A5, 342 m; A6, 339 m |
| 3 | G080095E42328N | - | Trunk glacier: 935 m |
| 4 | G079749E42092N | - | Tributary glacier: A1, 1118 m; A2, 655 m; A3, 672 m; A4, 495 m |
| 5 | G079839E41935N | Ayilangsu Glacier | Tributary glacier: A1, 637 m; A2, 1312 m; A5, 133 m; Trunk glacier: A3, 526 m; A4, 589 m |
| 6 | G079925E41780N | Bingtan Glacier | Trunk glacier: A1, 605m; A2, 2837m |
| 7 | G079990E41931N | Tuomuer Glacier | Trunk glacier: A2, 1049 m; A3, 3327 m; tributary glacier: A1, 1211 m; A4, 2807 m; A5, 284 m; A6, 2119 m |
| 8 | G080154E41960N | West Qiongtailan Glacier | Trunk glacier: A1, 2685 m; A2, 2056 m; tributary glacier: A3, 1210 m; A4, 2396 m |
| 9 | G080251E42005N | Qiongtailan Glacier | Tributary glacier: A1, 1684 m; trunk glacier: A2, 1584 m |
| 10 | G080352E42043N | Keqiketieliekesu Glacier | Trunk glacier: 1029 m |
| 11 | G080581E42004N | Qiongkuziwayi Glacier | Trunk glacier: 1040 m |
| 12 | G080949E42400N | Aerqialeteer Glacier | Tributary glacier: 512 m |
| 13 | G080922E42305N | Muzhaerte Glacier | Tributary glaciers: A1, 1979 m; A3, 1715 m; trunk glacier: A2, 1706 m |
| 14 | G081078E42318N | - | Trunk glacier: 2547 m |

## 4. Discussion

### 4.1. The Relationship between Advancing, Surging Glaciers, and Climate Change in the Tien Shan

Temperature and precipitation affect glacier ablation and accumulation, respectively, and the combination determines the nature, development, and evolution of a glacier [62]. Scholars have previously studied the relationship between advancing, surging glaciers, and climate change. The authors of [63] believed that repetition intervals of surges of the Variegated Glacier in Alaska are associated with variations in snow cover. The authors of [64] argued that the surge cycle of the Eyjabakkajökull Glacier is attributed to the mass balance driven by climate. The authors of [65] found that the surge repetition interval of the Donjek Glacier in Yukon, Canada, is influenced by the interaction of external (climate) and internal glacial processes, whereas the former has a more direct effect on surges. The authors of [66] revealed the correlation between glacial advance and anomalies of climate change in the Bukatage Mountains.

In this study, the ERA5-Land Climate Reanalysis dataset of the monthly temperature of air at 2 m above the surface of land and total precipitation with a spatial resolution of $0.1° \times 0.1°$ was selected to explore the relationship between the advance of glaciers and climate change in the Tien Shan Range. From the perspective of the interannual variation in temperature during the ablation period and annual precipitation (Figure 13), the temperature during the ablation period in the Tien Shan Range showed a rapidly increasing trend year by year, while the precipitation showed a moderate increase in the mountains and a yearly decrease in the river valleys. Among them, the eastern portion of the Central Tien Shan and the Eastern Tien Shan had the fastest warming rates (0.28–0.65 °C/10a), whereas the western portion of the Central Tien Shan and Western Tien Shan Mountains had relatively slow warming rates (0.05–0.35 °C/10a). Although precipitation is increasing in some areas, such as in the Bogdar and Eren Habirga mountains in the Eastern Tien Shan and Narat Mountain in the Central Tien Shan, a slight increase in precipitation has far less impact on glaciers than a rapid rise in temperature. Therefore, the glaciers in eastern Central Tien Shan and Eastern Tien Shan were rapidly retreating [39], whereas the Western Tien Shan and the western portion of the Central Tien Shan were areas where surging and advancing glaciers were concentrated. In terms of the major years (1997, 2000, 2002, 2013, 2015) and months (July, August, September) when glacial advance occurred (Figure 14), the temperature in Western and Central Tien Shan showed a steady and slow rising trend, whereas precipitation showed an increasing trend; however, both temperature and precipitation were decreasing in some individual years. Considering the lagged response of glacial changes to climate change [67], temperature and precipitation in the early years of this period showed an increasing and upward trend. In the Tien Shan Range, precipitation is mainly concentrated in summer, which is also the period of glacial melt in the region. The abnormal increase in precipitation leads to an increase in mass accumulation, which may break the mass balance of the glacier. Recently, rapid warming has accelerated the melting of glaciers, and the glacial meltwater may enter the ice bed through the hydraulic system within the glacier, resulting in reduced friction at the bottom of the glacier. Therefore, the glacial advance in the Tien Shan region is correlated with climate change, but it is difficult to explain the advance mechanism by climate change alone.

### 4.2. Periodicity of Surging Glaciers in the Tien Shan

The surge cycle refers to the time span between two surges, which includes both surge and quiescent phases [34]. The surge cycle of different glaciers varied greatly, but the same glacier experience relatively constant surge and quiescent phases [68]. The surge cycle of an individual glacier is usually several years to decades, or longer [66]. The Northern Inylchek Glacier in the middle of the Tien Shan Range reached its largest size in 1943, but continued to retreat from 1967 to 1992, with the glacier terminus in a stable state from 1992 to 1995 [69,70], after which the glacier began to retreat after a surge from 1995 to 1997. Therefore, the glacier's surge phases last about 2 years, and the quiescent phases are

about 50 years, with a surge cycle exceeding 50 years. The Samoilowich Glacier was in a retreat from 1960 to 1992. The maximum and minimum lengths of this glacier were about 8.9 km in 1960 and 5.8 km in 1992, respectively, retreating about one-third of the glacier's length [70,71]; then, it surged from 1999 to 2006. Therefore, the surge phases of the glacier last about 7 years and quiescent phases last about 39 years, with the glacier experiencing a surge cycle of about 50 years. The right tributary of the Shokalsky Glacier advanced in 1962–1964 [36], and the glacier advanced again in 1993–1996; therefore, the surge phases of this glacier last 2–3 years, the quiescent period lasts about 30 years, and the surge cycle lasts about 35 years. The surge phases of Mushketov [72,73] and Karagul [74] glaciers were 1–2 years, whereas the South Jangyryk Glacier and the glacier coded G079799E41948N were in surge phases for 4–7 years; the Bezymyanny Glacier [72,73], Bogatyr Glacier [36], and the glacier coded G080499E41872N had surge phases of 14–19 years. The surge cycles for other glaciers are unavailable due to a lack of pre-glacial advance information. Therefore, according to existing research data combined with this study, it was determined that the cycle of surge glaciers in the Tien Shan is roughly 35–60 years, the surge phases are variable, ranging from 1 to 19 years, and the quiescent phases last 30–50 years.

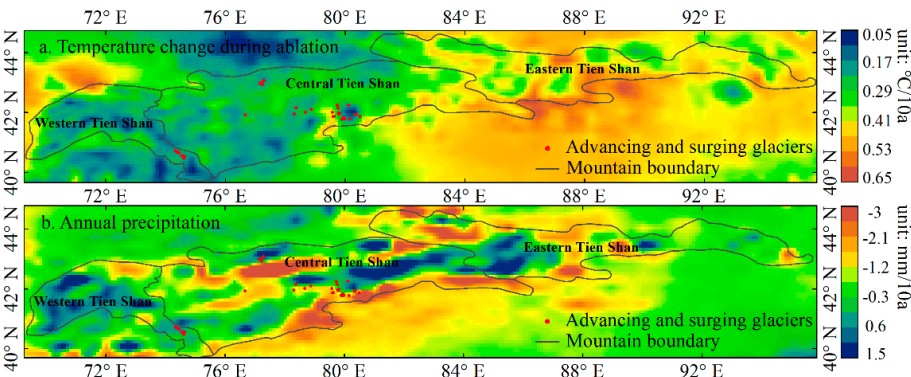

**Figure 13.** Heat maps show (**a**) the change in temperature during ablation and (**b**) annual precipitation in the Tien Shan Range from 1990 to 2019 (data from ERA5-Land Climate Reanalysis dataset).

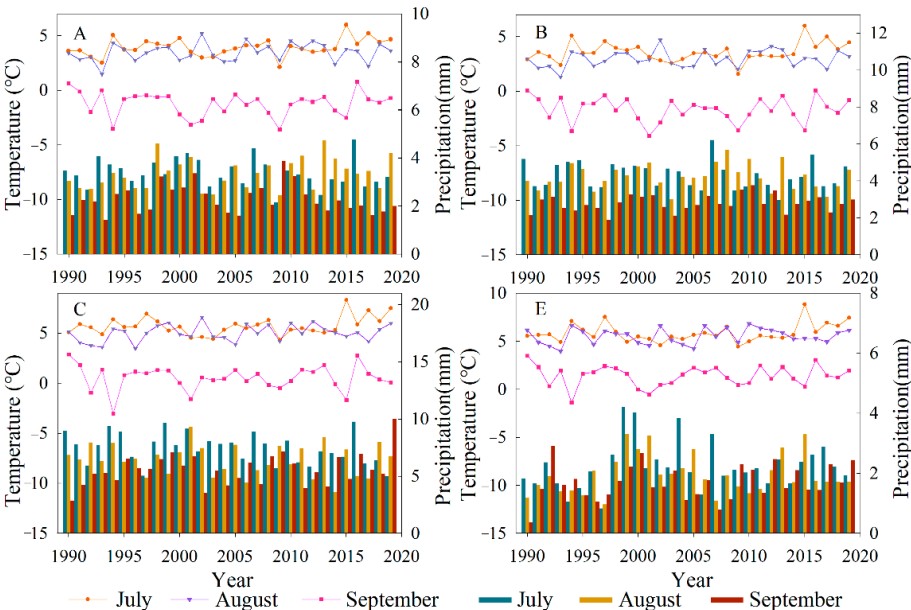

**Figure 14.** Change of temperature and precipitation in July, August, and September of glacial advance in the Tien Shan (the line graph shows the temperature, and the bar graph shows the precipitation; (**A–C,E**) in the figure correspond with locations in Figure 4; data from ERA5-Land Climate Reanalysis dataset).

*4.3. The Occurrence Mechanism of Surging Glaciers in the Tien Shan*

At present, the instability of glacial dynamics is generally believed to be the basic reason for glacial surges [75–77]. The authors of [78] argued that the instabilities of stress, temperature, and water film combine to promote glacial surges, and stress instability is the major factor. The authors of [79] considered that stress instability manifests itself as a change in a portion of the glacier from an elongated flow to a compressed flow. The authors of [10] consider that at the boundary between surge ice and dead ice, when the shearing stress reaches a certain critical value, a glacier begins to surge, causing the glacier to move downstream rapidly. In addition, the glacial meltwater is also regarded as a main cause of glacial surge. For example, on 20 September 2002, a surge event of the Colka Glacier in the Caucasus Mountains was regarded as caused by an excessive watering of the ice bed and ice interior [80]. The current explanations of surge glaciers mainly include thermal and hydrological control mechanisms [81–83]. The former deems that a change in the temperature field at the bottom of the glacier leads to deformation and an increase in the porosity of subglacial sediments, which triggers a glacial surge. However, the latter deems that a change in the drainage system at the bottom of the glacier from centralized to decentralized is the main driving factor that triggers a glacial surge. The ultimate reason for the two mechanisms is the increase in water pressure under the glacier, which enhances the sliding of the glacier along the bottom.

As far as the Tien Shan region is concerned, the influence of topography, availability of remote sensing images, and the difficulty of fieldwork have meant that no detailed study of the surge mechanism has been completed. Some research has been conducted on the Northern Inylchek Glacier. For example, the authors of [35] believed that the glacier surge characteristics are similar to those of the Karakoram and Alaskan regions. The reason is that the duration of a surge is relatively short (2 years), and the advance speed reaches about 50 m/d during the surge period, so the Northern Inylchek Glacier is considered to be of the Alaskan-type. A surge may be caused by the fragile deformation of the glacier during the surge period combined with lateral shear stresses exerted on the trunk glacier by the valley flanks, which reactivates a surge [35]. Moreover, a glacier with a relatively long ice tongue and a smaller slope can thicken more easily, store more water than other glaciers, and erode the glacier bed intensively [36], which is a possible reason for a surge. Both the Samoilowich and G080499E41872N glaciers had relatively short glacial snouts in the pre-surge period; their ice tongues advanced several kilometers during the surge. This is likely caused by the accumulation of mass in the upper reaches of the glacier, which caused the bottom temperature to reach the melting point of pressure, so that an increase in hydrostatic pressure at the bottom of the glacier led to rapid bottom sliding, which is similar to the characteristics of multi-thermal glaciers controlled by a hydrological mechanism [82]. There is a slower surge initiation of these two glaciers; the surge phase of the Samoilowich Glacier was about 5 years, reaching a maximum flow velocity before surge termination with a surge cycle of over 50 years; however, the G080499E41872N glacier had a surge phase as long as 19 years and its surge period is unclear currently, which is similar to the characteristics of a thermal glacier (controlled by a thermal mechanism) [81]. Differences exist in the initiation period, cycle, and mechanisms of different advancing or surging glaciers in the same part of the Tien Shan region. Combined with the previous descriptions, the advance of glacial termini in the Tien Shan region occurs in all seasons of a year, so we speculate that the glacier advance mechanism is influenced by a combination of thermal and hydrological mechanisms in the Tien Shan, as well as by climatic factors. Glacial advancing or surging is related to glacial meltwater, glacier internal structure, the degree of development of glacier surface fissures, and other factors. Clarifying the mechanism of glacial advancing or surging in the Tien Shan will require additional research and field investigations.

### 4.4. Comparison with Surging Glaciers in Other Regions of High Asia

In High Asia, surging glaciers mainly occurred in the Karakoram Range, Pamirs, and West Kunlun Mountains, which are known as "Karakoram anomalies" for their characteristics, such as the relative stability and advance of the glaciers in these regions. Among them, the Karakoram Range is one of the regions where surging glaciers are concentrated and have developed; the surge phases of glaciers in the Karakoram Range are relatively short, ranging from a few months to a few years [16,84]. Various researchers hold different views in the current research on the mechanism of the Karakoram glacial surge. Some argue that the Karakoram glacial surge is affected by thermodynamics, that is, the increased accumulation of ice at high elevations leads to the bottom of the glacier transforming into a compressed melt state. There are also observations and modeling of a single surge event, indicating that changes in hydrological conditions are the main cause of glacier surging [6]. In contrast, others argue that the emergence of surging and advancing glaciers in the Karakoram Range is largely dependent on the high elevation, complex topography, and climatic background [7,85].

The surge period of glaciers in the Pamirs ranges from several months to 10 years, with the surge period of most glaciers being more than 2 years. The surge period of the Bivachny Glacier in the West Pamir is about 4 years, that of 5Y663L0023 glacier in the East Pamir is about 4 years, and the Karayalak Glacier in the Gongger Nine Peaks is only several months. The surge cycle of the glaciers in the Pamirs is very short, with the Medvezhiy Glacial surge occurring every 10–14 years and the Bivachny glacier surging every 15–20 years. The reason for the timing of glacial surges in the Pamirs remains unclear. Some scholars believed that the Karayalak and 5Y663L0023 glaciers are more likely to be affected by thermodynamic mechanisms, but increased liquid precipitation and ice melt water are also important factors [11,56].

The active period of surge glaciers in the West Kunlun Mountains is more than 5 years, with a surge cycle of more than 42 years. The authors of [21] suggested that thermal and hydrological control mechanisms combined to cause glacial surging in the West Kunlun region. The authors of [86] pointed out that the temperature in summer in the West Kunlun region has risen significantly since the 1970s. The warming temperature has led to an increase in glacial meltwater, with large amounts of meltwater entering the ice and under the ice through fissures, thereby triggering glacial surging. The authors of [87] deemed that the ablation pressure and frictional thermal generation process can also cause the glaciers in the West Kunlun Mountains to produce subglacial meltwater in cold and dry environments, resulting in glacier surges.

In terms of the surge phases, quiescent phases, the month of occurrence, and the mechanism of a surge (Figure 15a–c), the surging glaciers in the Tien Shan Range are different from those in other regions of High Asia. Compared with those in other regions of High Asia, the surge glaciers in the Tien Shan Range have longer quiescent phases and surge cycles. The surge phase, from 1 to 19 years, is diverse. The quiescent phase is 30–50 years, or even longer, so that the surge cycle is roughly more than 50 years, and surges may occur during any season of the year. Therefore, the glacial surges in this region may be attributed to a combination of thermal and hydrological mechanisms and may also be influenced by factors, such as the geological type and friction at the bottom of a glacier.

### 4.5. Disaster Assessment of Surging or Advancing Glaciers in the Tien Shan Region

Surging glaciers can move several kilometers forward in a short period during a surge period, damaging vegetation, villages, roads, bridges, and other infrastructure along the way. In this study, we identified villages and roads within 5 km of the advancing and surging glaciers in the Tien Shan region, and found that the Kumubail and Achalayilak villages are relatively close to the glacier in the Tien Shan area and may be affected by the advancing glacier (Figure 16). Further study found that, although the village of Kumubail is only 1 km from glaciers coded G080154E41960N and G080251E42005N, both glaciers are surface advancing glaciers and have little direct impact on the village. Aqalyaylak village,

adjacent to the glacier coded G080499E41872N, is located on another side of the ridge so will barely be affected by the advance of this glacier. If the glacier coded G080499E41872N moves again, it may cause damage to vegetation and pastures along the roads. Therefore, the direct impact of advancing or surging glaciers on villages and roads in the Tien Shan region will be small or almost non-existent. However, related studies have found that surging glaciers led to glacial lake outbursts during advancing, thus triggering flood disasters in the Tien Shan region [35,61]. According to the authors of [35], the surging of Northern Inylchek glaciers led to the glacial lake (Upper Lake Merzbacher) outburst flood, and the lake water entered the Inylchek River and the Sary-Djaz River by flowing through the subglacial or glacial interior of the Southern Inylchek Glacier. Floods increase the flow of downstream rivers, thus posing a great threat to the safety of people's lives and properties in downstream coastal areas. Therefore, more attention should be paid to disasters such as glacial lake outburst floods due to glacier advancing and surging in the Tien Shan region, and remote sensing observations and fieldwork should be enhanced.

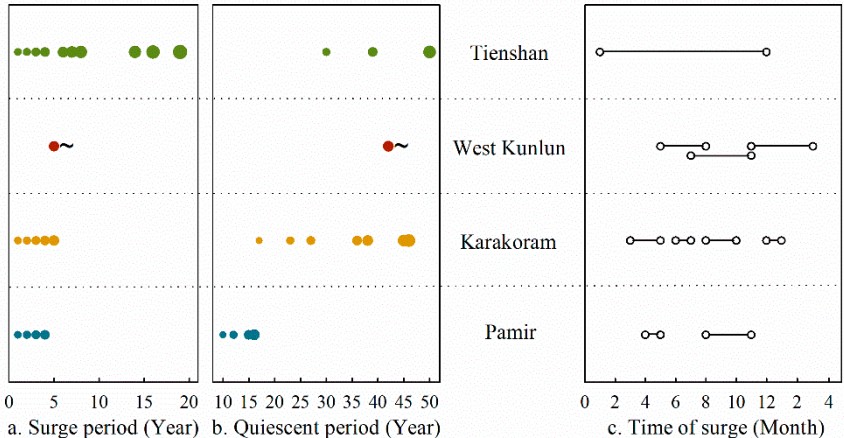

**Figure 15.** The surge phase (**a**), quiescent phase (**b**) and the time of surge (**c**) in different regions of High Asia (●~—indicates a year greater than that (5, 42) indicated by the symbol. Data in the figure are from the literature ([11,16,21,56,84,88,89]; this study)).

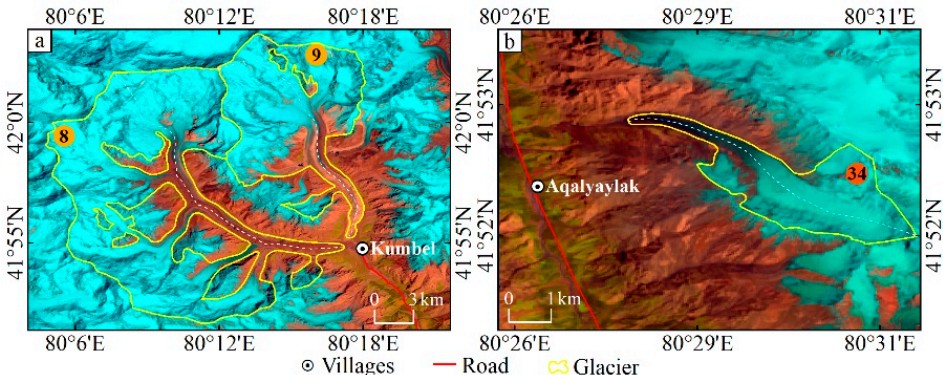

**Figure 16.** Hazard assessment for two advancing and surging glaciers near the villages of (**a**) Kumbel and (**b**) Aqalyaylak in the Tien Shan region.

## 5. Conclusions

Based on Landsat TM/ETM+/OLI remote sensing images, we identified 48 glaciers advancing from 1990 to 2019 in the Tien Shan Range. Advances occurred at the terminus of 34 glaciers, while 14 glaciers advanced on their surfaces. For the 34 glaciers, 10 were identified as surging glaciers and 24 were identified as advancing glaciers. These glaciers are distributed centrally in the western part of the Halik and Kungey Mountains in the Central Tien Shan, and Fergana Mountain in the Western Tien Shan.

From 1990 to 2019, a total of 169 advances occurred on 34 advancing and surging glaciers in the Tien Shan region, based on annually available remote sensing images. Among them, the right tributary of one advancing glacier, G074380E40695N, and a surging glacier, G080499E41872N, have advanced the greatest number of times (10 and 18 times, respectively); glacial advance occurred mainly in 2013 (10 times), mainly in July (26 times) and August (23 times). The advances of surging glaciers were concentrated in 2000 (six times) and 2002 (six times), with the months of surge concentrated in July (14 times).

The cycle of the surge glaciers in the Tien Shan region is 35–60 years, and the surge phases are variable, ranging from 1 to 19 years, with quiescent periods of 30–50 years. Compared with other regions in High Asia, the surge cycle, surge phases, and quiescent phases are all longer in the Tien Shan. Surging glaciers in the Tien Shan region may be controlled by a combination of thermal and hydrological conditions; an increase in temperature and increased precipitation plays a driving role on the occurrence of surging glaciers, but it is still difficult to explain the mechanism of change based on a single climatic change. In the future, observations of climatic elements and glacier mass balance need to be strengthened. Remote sensing monitoring and fieldwork on elements, such as the ice surface velocity, the channel of water system within the ice and the crevasse development degree of advancing glaciers in the Tien Shan region, will be carried out to further clarify the happening mechanism and triggering mechanism of advancing glaciers.

**Author Contributions:** X.Y. designed the overall research plan of this study and revised the manuscript. S.Z. completed the identification of advancing and surging glaciers in the study area and wrote the first draft. D.Z. devised an algorithm for automatic extraction of glacier centerlines. Y.Z. completed the extraction of some advancing and surging glacier boundaries. S.L. and Y.M. reviewed and revised the manuscript. All authors played a role in review and editing this paper. All authors have read and agreed to the published version of the manuscript.

**Funding:** This research was funded by the National Natural Science Foundation of China (grant numbers: 41861013, 42071089); Open Research Fund of National Earth Observation Data Center of China (NODAOP2020007); Open Research Fund of National Cryosphere Desert Data Center of China (20D02); Northwest Normal University Graduate Student Research Grant Program (2019KYZZ012054).

**Conflicts of Interest:** The authors declare no conflict of interest.

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
