# Peer review of "Remote Sensing Monitoring of Advancing and Surging Glaciers in the Tien Shan, 1990–2019"

_remotesensing, doi:10.3390/rs13101973_

Round 1

Reviewer 1 Report

Congratulations for the good work.

Author Response

Thanks for your approval of this study.

Reviewer 2 Report

please provide the uncertainty range for glacier terminus advance in tables 1 and 2 as well as the main text.

Author Response

Thanks for your suggestions. In section 2.3.2 of the manuscript, we have evaluated the accuracy of glacier length extraction. In this study, we consider that the accuracy of glacier length depends on the centerline, which is affected by the accuracy of glacier outline and the quality of the DEM data. Among them, the DEM data only affect the selection of the local highest and the lowest points of the glacier outline, and its effect on the accuracy of glacier length can be ignored. Thus, the accuracy of glacier length relies on the spatial resolution of remote sensing images used and the accuracy of glacier outline. When assuming the glacier outline is correct, the absolute error of each glacier length is the size of a single pixel of Landsat TM/ETM+/OLI image, i.e., 30 m/30 m/15 m. Therefore, we used relative error to assess the accuracy of glacier length in this study.

This manuscript is a resubmission of an earlier submission. The following is a list of the peer review reports and author responses from that submission.

Round 1

Reviewer 1 Report

This study monitoring the glacier and glacier change using multi-temporal Landsat Images in the Tien Shan during 1990–2019. Glacier change (such as advancing terminus position and surge cycle as well as surface expending) was identified based on the conventional threshold method with additional manual improvement and glacier length extraction. Driving factors such as temperature, precipitation are also analyzing to explain the glacier variation in the region. However, I have a major concern about the quality of glacier mapping due to the current version of the study did not provide an accurate assessment.

Comments:

  • Please provide assessthe accuracy of glacier outlines! Are all threshold values same for all images for clean glacier mapping! Please provide the sensitivity analysis for the different threshold values.
  • At least, selected several glaciers (clean and debris-covered) performed the round-robin experiment with a focus on comparing glacier outlines digitized manually on satellite images with different resolutions and from automated methods (Paul et al. 2013).
  • Surface advancing glaciers- from Fig 6-7 show that images with cloud cover and seasonal snow cover were used for the extraction of glacier outlines. It may result in unreal changes in the glacier area, especially in the accumulation area.
  • Information (separate analysis of clean and debris-covered glaciers in advancing and area change as well as their variation in snow accumulation area ratio ) may make this study more interesting to the  

Other:

Line16  Thirty-four glaciers have exhib- 16 ited terminal advances, with14 glaciers advancing on the glacial surface

Please replace the words advancing, maybe expand, or other

Line17-18  Ten of the 34 glaciers experiencing terminal advances have been identified as surging glaciers, whereas the others were advancing glaciers.

Please check consistency in number formatting, 34 or Thirty-four(line 16)

Line21-23  From 1990 to 2019, a total of 169 advances occurred on 34 terminal advancing glaciers in the Tien Shan region, with the most advancing and surging occurring in July (26 and 14 times, respectively), the most 23 surging in 2000 (10 times), and the most advances in 2013 (14 times).

Please improve these sentences. What is the meaning of “times”!

Line118-120 A total of 702 remotely sensed scenes were used in this study (Figure 2), in- 118 cluding Level-1 products of the Landsat Thematic Mapper (TM), Enhanced Thematic 119 Mapper (ETM+), and Operational Land Imager (OLI) downloaded from the United States 120 Geological Survey (https://earthexplorer.usgs.gov).

How did you select these Landsat images and what is the condition of selection (time, cloud cover, snow cover)

Line121-123 All the images had been treated with radiation, geometric, and topographical corrections based on digital elevation mode (DEM) data.

Are these corrections done by the author or already done by the data provider?

Reviewer 2 Report

Dear Authors,
Many thanks for this extremely detailed analysis of glaciers in Tien Shah. I really enjoyed going through it, and can appreciate the amount of work that has gone into the research work and the manuscript itself. Overall, I think your paper is a useful and valuable addition to the scientific community. My criticism, apart from some minor details, is more directed to the fact that the paper is extremely descriptive, and maybe some details can be better summarized in for example bar or histogram plots. I would also like to suggest, and recommend that the discussion section is modified to have less descriptive elements, and a better focus on the process dynamics - perhaps, a conceptual model of the surge dynamics would be a good addition. This would also help to compare with glacier types, processes, and dynamics from other regions (which is something that you already mention). Finally, the risk or disaster assessment part is rather weak (i.e. not well-supported). Since this is one of the key objectives you mention, it would be important to give it some weight for example in the results section too. 

In summary, I think this paper covers extremely well the first two of your main objectives. However, the dynamics and links to climate change objective requires some additional work, mainly in the discussion section, along with the question of assessment of potential risks that these systems pose. 

Please note that I have attached a pdf with my quick notes and comments. Highlight sections need to be checked for grammar or are just words / phrases that appeared odd to me.

Finally, I would encourage you to take my suggestions for improvement as a challenge to make this an even better piece of work !

with regards

Reviewer 3 Report

The manuscript is an attempt to address a the topic of surging glaciers with satellite observation. The paper however lacks sufficient ground validation and discussion & comparison of the available methods to remotely monitor the surging glaciers with satellite images. The authors are requested to resubmit the manuscript after a detailed change in the document pertaining to validation and methodical nuances.

Reviewer 4 Report

I am not convinced by the authors, and the paper lacks scientific approach and soundness. A clear definition of a surging glacier is given, but is not used in the subsequent analysis. Hence, the data in Discussionon surge timing, periods and their duration , e.g. on the comparison figure, is highly likely invalid, like it is in the Table 3. Some surges are only identified using visual properties of the glacier surface, but not supported by ice flow rate data - how this can be acceptable ?! Methods are poorly described, and description is sometimes irrelevant to the terminology used - e.g., the authors discuss surface elevation, while the surface elevation data sources were not presented in the Data sources section. There are plenty of comments concerning the methodological side of the manuscript, and the flaws present therein, they are presedted in an attached file.

Numerous sources are missing that could put this paper in a proper context. Comparison with other papers show that mass gain is observed in numerous glaciers in the Tien Shan region, and within certain altitude limits, e.g. in Aksu R. basin (Baojuan et al., 2017). Could the authors better relate to these already available data in their discussions?
